# Manifestations of Liver Impairment and the Effects of MH-76, a Non-Quinazoline α1-Adrenoceptor Antagonist, and Prazosin on Liver Tissue in Fructose-Induced Metabolic Syndrome

**DOI:** 10.3390/metabo13111130

**Published:** 2023-11-03

**Authors:** Monika Kubacka, Barbara Nowak, Monika Zadrożna, Małgorzata Szafarz, Gniewomir Latacz, Henryk Marona, Jacek Sapa, Szczepan Mogilski, Marek Bednarski, Magdalena Kotańska

**Affiliations:** 1Department of Pharmacodynamics, Faculty of Pharmacy, Medical College, Jagiellonian University, Medyczna 9, 30-688 Krakow, Poland; monika.kubacka@uj.edu.pl (M.K.); jacek.sapa@uj.edu.pl (J.S.); szczepan.mogilski@uj.edu.pl (S.M.); 2Department of Cytobiology, Faculty of Pharmacy, Medical College, Jagiellonian University, Medyczna 9, 30-688 Krakow, Poland; barbara.anna.nowak@uj.edu.pl (B.N.); monika.zadrozna@uj.edu.pl (M.Z.); 3Department of Pharmacokinetics and Physical Pharmacy, Faculty of Pharmacy, Medical College, Jagiellonian University, Medyczna 9, 30-688 Krakow, Poland; malgorzata.szafarz@uj.edu.pl; 4Department of Technology and Biotechnology of Drugs, Faculty of Pharmacy, Medical College, Jagiellonian University, Medyczna 9, 30-688 Krakow, Poland; gniewomir.latacz@uj.edu.pl; 5Department of Bioorganic Chemistry, Faculty of Pharmacy, Medical College, Jagiellonian University, Medyczna 9, 30-688 Krakow, Poland; henryk.marona@uj.edu.pl; 6Department of Pharmacological Screening, Faculty of Pharmacy, Medical College, Jagiellonian University, Medyczna 9, 30-688 Krakow, Poland; marek.bednarski@uj.edu.pl

**Keywords:** metabolic dysfunction-associated fatty liver disease, hypertension, α1-adrenoceptor antagonist, fructose, metabolic syndrome

## Abstract

Excessive fructose consumption may lead to metabolic syndrome, metabolic dysfunction-associated fatty liver disease (MAFLD) and hypertension. α1-adrenoceptors antagonists are antihypertensive agents that exert mild beneficial effects on the metabolic profile in hypertensive patients. However, they are no longer used as a first-line therapy for hypertension based on Antihypertensive and Lipid-Lowering Treatment to Prevent Heart Attack Trial (ALLHAT) outcomes. Later studies have shown that quinazoline-based α1-adrenolytics (prazosin, doxazosin) induce apoptosis; however, this effect was independent of α1-adrenoceptor blockade and was associated with the presence of quinazoline moiety. Recent studies showed that α1-adrenoceptors antagonists may reduce mortality in COVID-19 patients due to anti-inflammatory properties. MH-76 (1-[3-(2,6-dimethylphenoxy)propyl]-4-(2-methoxyphenyl)piperazine hydrochloride)) is a non-quinazoline α1-adrenoceptor antagonist which, in fructose-fed rats, exerted anti-inflammatory, antihypertensive properties and reduced insulin resistance and visceral adiposity. In this study, we aimed to evaluate the effect of fructose consumption and treatment with α1-adrenoceptor antagonists of different classes (MH-76 and prazosin) on liver tissue of fructose-fed rats. Livers were collected from four groups (Control, Fructose, Fructose + MH-76 and Fructose + Prazosin) and subjected to biochemical and histopathological studies. Both α1-adrenolytics reduced macrovesicular steatosis and triglycerides content of liver tissue and improved its antioxidant capacity. Treatment with MH-76, contrary to prazosin, reduced leucocytes infiltration as well as decreased elevated IL-6 and leptin concentrations. Moreover, the MH-76 hepatotoxicity in hepatoma HepG2 cells was less than that of prazosin. The use of α1-adrenolytics with anti-inflammatory properties may be an interesting option for treatment of hypertension with metabolic complications.

## 1. Introduction

Consumption of fructose has increased dramatically over the last decades, mainly due to the high intake of industrially manufactured food, in which sweeteners like high fructose glucose syrup are widely used. Excessive fructose consumption has been associated with pathogenesis of the metabolic syndrome and may be regarded as one of the causes of visceral obesity and insulin resistance. A diet rich in fructose may also impair liver function, leading to non-alcoholic fatty liver disease (NAFLD) [1,2,3,4]. Recently, in 2023, the current nomenclature has been revised, and the term metabolic-associated fatty liver disease (MAFLD) has been proposed [5]. The definition is based on the presence of hepatic steatosis and at least one other condition such as obesity, type 2 diabetes mellitus, or at least two metabolic abnormalities: increased waist circumference, arterial hypertension, hypertriglyceridemia, low high-density cholesterol (HDL), insulin resistance and chronic low-grade inflammation [2]. MAFLD commonly develops with obesity and insulin resistance and is considered as the hepatic component of metabolic syndrome, with prevalence increasing worldwide. The vast majority of MAFLD patients are overweight or obese and meet the criteria of the metabolic syndrome [1,3,4]. 

MAFLD induces a broad spectrum of manifestation of fatty liver, ranging from simple steatosis, steatosis with inflammation, steatosis with hepatocyte injury, steatosis with sinusoidal fibrosis, and ultimately in the long term, development of fibrosis and cirrhosis with an increased risk of hepatocellular carcinoma [6,7,8,9].

In metabolic syndrome, the liver is affected not only by excess of fructose from food but also by an inflammatory adipocytokines from enlarged visceral adipose tissues [7,8]. Inflammation and oxidative stress can activate pro-inflammatory kinases, induce inflammatory factors such as IL-1β, TNF-α, IL-6 and MCP-1 (monocyte chemotactic protein-1) [10] and impair insulin signaling in both hepatic and adipose tissues [11].

MAFLD, coexisting with insulin resistance and inflamed adipose tissue is responsible for a plasma environment conducive to atherosclerosis, with increased concentrations of triglycerides, glucose and insulin. In MAFLD hepatocytes produce and release a number of chemical compounds that create an atherogenic environment, hypertension and increased blood clotting. Moreover, premature and increased cardiovascular mortality is observed in patients with MAFLD [2,12].

There is also compelling evidence from animal and human studies suggesting that overactivation of the sympathetic nervous system is a key contributor to the development of MAFLD [6]. The sustained sympatho-excitation in obesity not only contributes to the rise in blood pressure but also facilitates further weight gain and progression of associated co-morbidities [13]. There is also evidence of the strong involvement of the sympathetic nervous system in the pathogenesis of liver diseases, especially liver fibrosis. Norepinephrine was reported to exert strong profibrogenic effect in the liver [14]. Norepinephrine, acting at α1-adrenoceptors, induces hepatic stellate cell (HSC) proliferation and increases the expression of collagen-1 [14,15,16].

α1-adrenoceptor antagonists are antihypertensive drugs, which, in addition to lowering blood pressure, have been shown to have mild beneficial effects on the metabolic profiles of patients with hypertension, modulating insulin activity and lipoprotein metabolism [17,18]. α1-adrenoceptor antagonists, through vasodilatation of blood vessels, improve regional blood flow and glucose delivery to the skeletal muscles. Moreover, α1-adrenergic stimulation inhibits insulin release and activates hepatic glucose production by enhancing glycogenolysis [19]. Therefore, blockade of α1-adrenoceptors may exert beneficial metabolic effects. However, in the Antihypertensive and Lipid-Lowering Treatment to Prevent Heart Attack Trial (ALLHAT) study, a quinazoline-based α1-adrenoceptors antagonist doxazosin was associated with higher risk of combined cardiovascular disease events, and since then α1-adrenolytics have no longer been used as a first-line therapy for hypertension [17]. There are also studies indicating that quinazoline-based α1-adrenoceptors antagonists induce the apoptosis and necrosis of cardiomyocytes, which may partially explain the ALLHAT outcomes [20,21]. Moreover, pre-clinical and clinical studies implicate a potential value of analogues of quinazoline-based α1-adrenoreceptor antagonists in prostate and other cancer types’ prevention and therapy due to induction of apoptosis pathways and cytotoxic properties [22,23]. The apoptotic properties of α1-adrenoceptors antagonists have been proved to be independent of α1-adrenoceptor blockade and are related to the presence of quinazoline moiety [21]. Moreover, apoptosis induction was never shown for non-quinazoline α1-adrenolytics such as urapidil or tamsulozine [20,21,24].

There is also much evidence from preclinical studies that α1-adrenoceptor antagonists may exert hepatoprotective effects; it has been shown that blockade of α1-adrenoceptors with doxazosin reduced collagen type I deposits and TGF-β-secreting cells in rodent models of carbon tetrachloride (CCl4)-induced cirrhosis [16]. Similar results were found in a rat model of alcohol fatty liver disease, where carvedilol, an α1- and β-adrenoceptor antagonist, attenuated hepatosteatosis, reduced the activation of HSCs, and decreased the deposition of collagen [25]. These effects may be attributed at least partially to blockade of α1-adrenoceptors, as prazosin, an α1-adrenoceptor antagonist, has been shown to reduce liver injury in a mouse model of NASH, whereas propranolol, a β-adrenoceptor antagonist, was not active or even enhanced liver injury [26]. Moreover, prazosin and other α1-adrenoceptor antagonists prevented paracetamol-induced hepatotoxicity in mice, probably due to prevention of paracetamol-induced microvascular disturbances, such as reduced sinusoidal perfusion and congestion [27]. Furthermore, there are studies showing that α1-adrenoceptor antagonists may reduce mortality in COVID-19 patients due to anti-inflammatory properties [28].

Chronic fructose-fed rats provide a useful experimental model for studying the interaction of the factors that shape metabolic syndrome [29]. In our previous study, we assessed the influence of non-quinazoline α1-adrenoceptor antagonist MH-76 (1-[3-(2,6-dimethylphenoxy)propyl]-4-(2-methoxyphenyl)piperazine hydrochloride), (Figure 1) as well as prazosin, a quinazoline-based α1-adrenoceptor antagonist, on metabolic syndrome generated by high fructose diet in rats [30]. MH-76 does not contain quinazoline moiety in its structure (Figure 1) and is a reversible and competitive antagonist of α1-adrenoceptors, with no selectivity for a specific α1-adrenoceptor subtype. In addition to its antihypertensive effect, MH-76 decreased hyperglycemia and hypertriglyceridemia, and, contrary to prazosin, prevented abdominal obesity and reduced insulin resistance. Moreover, MH-76 exerted anti-inflammatory properties in adipose tissue by reducing pro-inflammatory cytokine production and inhibiting inflammatory cells’ recruitment [30,31].

In this study, we aimed to evaluate the effect of fructose consumption and treatment with α1-adrenoceptor antagonists of different classes on liver tissue in detail. We also compared the effects of a non-quinazoline α1-adrenoceptor antagonist, MH-76, and prazosin on some of the critical points associated with pro-inflammatory state in hepatic tissue of fructose-fed rats, a well-validated model of metabolic syndrome.

## 2. Materials and Methods

### 2.1. Experimental Protocol

Livers were collected from fructose-fed rats with metabolic syndrome from our previous experiment [30]. All experimental procedures were conducted in accordance with the guidelines of the National Institutes of Health Animal Care and Use Committee and approved by the Local Ethics Committee on Animal Experimentation (resolutions no. 338/2017 and 187/2018) in Krakow, Poland. Male Wistar rats (Krf: (WI) WU) weighing 190–210 g, age 7 weeks, obtained from an accredited animal house at the Faculty of Pharmacy, Jagiellonian University Medical College, Krakow, Poland, were used. Rats were administered with 20% fructose solution instead of drinking water for 18 weeks ad libitum, whereas the control group was maintained on normal drinking water for 18 weeks. To study the effects of the test compounds (MH-76, prazosin) they were administered during the last 6 weeks of the 18-week experiment in fructose-fed rats. Rats were randomly divided into 4 groups as follows: 

Control (*n* = 8): Animals received regular diet and water ad libitum for 18 weeks. After 12 weeks, this group received saline (1 mL/kg intraperitoneally (i.p.) daily) during the last 6 weeks of the experiment. 

Fructose (*n* = 8): Animals received a regular diet and fructose was administered as 20% solution in drinking water for 18 weeks. After 12 weeks, this group received saline (1 mL/kg i.p. daily) during the last 6 weeks of the experiment.

Fructose + MH-76 (*n* = 8): Animals received a regular diet and fructose was administered as 20% solution in drinking water for 18 weeks. After 12 weeks, this group received MH-76 5 mg/kg/day i.p. during the last 6 weeks of the experiment.

Fructose + Prazosin (*n* = 8): Animals received a regular diet and fructose was administered as 20% solution in drinking water for 18 weeks. After 12 weeks, this group received prazosin 0.2 mg/kg/day i.p. during the last 6 weeks of the experiment [30,31]. Figure 1 shows timeline of the experiment.

The systolic blood pressure was measured once a week, on weeks 0, 1–18, at the same time of the day. Body weight and fasting glucose concentration were measured at the baseline and throughout the study [30].

At the end of the experiment, after 16 h fasting but with free access to water, all rats were anesthetized with thiopental (75 mg/kg i.p.) and decapitated. Blood was collected for biochemical assays (glucose, insulin, total cholesterol, HDL cholesterol, triglycerides, uric acid). Insulin resistance was assessed using HOMA-IR index [30]. Livers were dissected out and weighed by investigators unaware of the groups’ allocation. Livers were stored at −80 °C until assayed.

### 2.2. Drugs and Chemicals

Prazosin (prazosin hydrochloride) was purchased from Tocris, Bristol, UK. Compound MH-76 was synthesized in the Department of Bioorganic Chemistry, Chair of Organic Chemistry, Pharmaceutical Faculty, Jagiellonian University [32], Figure 1. 5,5′-dithio-bis-2-nitrobenzoic acid (DTNB), glutathione reduced form (GSH), 1,1′,3,3′-tetraethoxypropane (TEP), tetrachloroacetic acid (TCA) and thiobarbituric acid (TBA), 2,4,6-tripyridyl-s-triazine (TPTZ) were obtained from Merck, Sigma-Aldrich, Darmstadt, Germany. HCl and FeCl_3_ × 6H_2_O were obtained from POCh, Gliwice, Poland.

### 2.3. Biochemical Assays

#### 2.3.1. Preparation of Tissue Homogenates

Frozen livers were weighed, and homogenates were prepared by homogenization of 1 g of the tissue in 4 mL of 0.1 M phosphate buffer, pH 7.4, using the IKA-ULTRA-TURRAX T8 homogenizer. The obtained homogenate was centrifuged at 1500× *g* for 10 min and the supernatant was used for biochemical assays. All experimental procedures involved in the preparation of tissue homogenates were carried out at 4 °C.

The concentrations of TNF-α (E0764Ra, Bioassay Technology Laboratory, Birmingham, UK), IL-6 (E0135Ra, Bioassay Technology Laboratory) in livers were determined by the ELISA method with the use of commercially available kits. The concentrations of leptine, MCP-1, PAI-1 and IL-1β were determined using a commercially available MilliplexTM MAP Kit (RADPCMAG-82K, Millipore, Billerica, MA, USA) according to the manufacturer’s protocol. The quantitative analysis was performed using a MAGPIX Luminex analyzer with xPONENT software 3.1. (Luminex Corporation, Austin, TX, USA).

Levels of analytes were calculated based on the standard curves using the spline curve-fitting method and were expressed in pg/mL or ng/mL of homogenate.

To determine total cholesterol or triglyceride levels in plasma, standard enzymatic, spectrophotometric tests (Biomaxima S.A., Lublin, Poland) were used [33]. The substrate was decomposed with appropriate enzymes for the relevant product, which was converted to a colored compound. Coloration was proportional to their concentration. The absorbance was measured at a wavelength of 500 nm.

#### 2.3.2. Determination of the Non-Protein Thiols Levels (NPSH)

Determination of NPSH levels is based on Ellman’s method. 5,5′-dithio-bis-2-nitrobenzoic acid (DTNB) is reduced by –SH group to 2-nitro-5-thiobenzoic acid (TNB) characterized by intensive yellow color, which shows maximum absorbance at 412 nm [34]. Briefly, 950 μL of the studied homogenate was first deproteinized by addition of 50 μL of cold 50% TCA, and then the sample was centrifuged at 10,000× *g* at a temperature of +4 °C for 10 min. To 850 μL of 0.2M phosphate buffer (pH 8.2), 100 μL of 6 mM DTNB and 50 μL of supernatant from deproteinized homogenate were added. Absorbance was measured at a wavelength λ = 412 nm 1 min after supernatant addition. The total content of NPSH was determined from a standard curve prepared for the 1 mM glutathione reduced form (GSH). 

#### 2.3.3. The Ferric Reducing Antioxidant Power (FRAP) Assay

The assay was performed according to Benzie and Strain with some modifications [35]. The total reducing activity of liver tissue after treatment with tested compounds was determined by spectrophotometric detection of the reduced iron concentration. In the experiment, 20 µL of homogenate was mixed with 180 µL of the reagent with the following composition: 10 parts of a 0.3 M sodium–acetate buffer, pH 3.7, 1 part of 0.01 M TPTZ solution and 1 part of 0.02 M FeCl_3_ × 6H_2_O solution. Absorbance was measured after 10 min incubation at room temperature at 593 nm. The results were presented as the amount of reduced iron (II) ions. FeSO_4_ × 7H_2_O salt was used for standard curve construction. For this purpose, the salt was dissolved in water and further diluted to obtain concentrations from 0.025 to 2 mM.

### 2.4. Histopathological Evaluation and Image Analysis and Identifying and Quantifying Liver Fibrosis

Liver specimens were quickly removed, immersed in 10% formalin, dehydrated and embedded in paraffin, sectioned at 5 μm, stained with hematoxylin and eosin (H&E), Goldner Masson Trichrome stain and Sirius red in consecutive slides, and evaluated by light microscopy (Olympus BX41, Evident, Tokyo, Japan) and color camera (Olympus UC90, Evident, Tokyo, Japan). The digital images (digitalized at 3384 × 2708 pixel resolution) were then analyzed using computerized imaging CellSensDimension v. 1.18 (Olympus, Tokyo, Japan) software. The histopathological scoring analysis was performed according to Kleiner et al. and Takahashi et al. [36,37]. The assessment was expressed as semi-quantitative score grades from 1 (minimal), 2 (mild), 3 (moderate), to 4 (marked) for each of the following parameters from liver sections: overall hepatic lobular structure (low power objective 4×), glycogen deposits, steatosis, periportal fibrosis and inflammation, perivenular (centrilobular) fibrosis and inflammation, sinusoidal changes and cellularity, and pigment deposits (presumably hemosiderosis) (high power objective, 40× and 100×). 

### 2.5. Hepatotoxicity Assay

The hepatotoxicity of prazosin and MH-76 was determined in HepG2 cell line (HB-8065™) obtained directly from American Type Culture Collection (ATCC^®^, Manassas, VA, USA). The cells were cultured according to the protocols provided by ATCC^®^. The hepatotoxicity assay procedure has been previously described [38]. Briefly, the cells were seeded to the 96-well plate 24 h before the assay (7000/well). The compounds were diluted in growth media and added to the cells in the final concentration range (0.1–100 µM). The reference cytostatic drug doxorubicin was added at 1 µM. The cells’ viability was determined after 72 h of incubation (37 °C, 5% CO_2_) by CellTiter 96^®^ AQueous Non-Radioactive Cell Proliferation Assay (Promega, Madison, WI, USA). The absorbance (at 490 nm) was measured using a microplate reader, EnSpire (PerkinElmer, Waltham, MA, USA). All the compounds were tested in quadruplicate in two independent experiments.

### 2.6. Statistical Analysis

Results are expressed as means ± SD or median ± inter-quartile range (IQR.) Statistically significant differences between groups were calculated using one-way ANOVA and the post-hoc Tukey multiple comparison test with all possible pairwise comparisons. The normality of data sets was determined using the Shapiro–Wilk test. For histopathological studies comparisons of evaluated parameters between the experimental groups were made using one-way analysis of variance (Kruskal–Wallis ANOVA). The comparisons among groups were performed using test by ranks followed by Dunn’s post-hoc test with all pairwise comparisons. Statistical analyses were performed using Graph Pad Prism 6.0 (Boston, MA, USA) and Statistica 13.3 software (StatSoft Polska, Krakow, Poland). Differences were considered significant at *p* < 0.05.

## 3. Results

### 3.1. Effect of MH-76 and Prazosin on the Concentration of Proinflammatory Factors in Liver Tissue

Fructose feeding significantly increased IL-6 concentration in liver tissue in all fructose-fed groups (Control: 56.60 ± 7.36, Fructose: 86.05 ± 8.12 pg/mL, (*p* < 0.0001), Fructose + MH-76: 70.43 ± 2.78 pg/mL, (*p* < 0.01), Fructose + Prazosin: 103.5 ± 8.11 pg/mL (*p* < 0.0001)). Treatment with MH-76 decreased IL-6 level compared with the Fructose group: in liver tissue from Fructose + MH-76 rats the concentration of IL-6 was markedly lower than in Fructose (Fructose: 86.05 ± 8.12 pg/mL, Fructose + MH-76: 70.43 ± 2.78 pg/mL, *p* < 0.01). Moreover, the concentration of IL-6 in livers from Fructose + Prazosin rats was significantly higher than from the Fructose (*p* < 0.01) and Fructose + MH-76 groups (*p* < 0.0001), (Figure 2a).

Fructose consumption had no effect on monocyte chemoattractant protein 1 (MCP-1), interleukin-1β (IL-1β), and TNF-α concentration in liver tissue. There were no statistically significant differences among experimental groups (Figure 2b–d). Similarly, fructose feeding did not influence PAI-1 concentration in liver tissue (Figure 2e).

### 3.2. Effect of MH-76 and Prazosin on Non-Proteine Thiols and Total Reducing Activity in Liver Tissue

Fructose feeding caused a significant decrease in NPSH in liver tissue (Control: 0.64 ± 0.12 pmol/mL, Fructose: 0.36 ± 0.11 pmol/mL, *p* < 0.05). Treatment with both MH-76 and prazosin tended to increase the NPSH concentration in liver tissue (there were no statistically significant differences between the NPSH levels in the livers of rats from the MH-76 or prazosin-treated groups and those of control rats) (Figure 3a).

Fructose feeding caused a significant (by ca. 15%) decrease in total reducing activity in liver tissue. Treatment with both, MH-76 and prazosin significantly increased reducing power in liver tissue by 6 and 5%, respectively (Figure 3b).

### 3.3. Effect of MH-76 and Prazosin on the Lipid Content and Leptin Concentration in Liver Tissue

Fructose feeding caused a marked increase in tryglicerydes content in liver tissue (Control: 1.37 ± 0.27 mmol/L, Fructose: 2.29 ± 0.76 mmol/L, *p* < 0.05). Treatment with both, MH-76 and prazosin tended to decrease the elevated triglycerides concentration in liver tissue (Figure 4a). 

Fructose consumption had no effect on total cholesterol content in liver tissue. There were no statistically significant differences among experimental groups (Figure 4b).

Fructose feeding resulted in increased leptin level in rats’ liver tissue, compared with the values observed in rats fed the standard diet (24.23 ±3.07 vs. 48.11 ± 5.68 pg/mL, *p* < 0.01, Figure 4c). Treatment with MH-76 tended to decrease the elevated leptin level (48.11 ± 5.68 vs. 36.94 ± 3.22 pg/mL), whereas prazosin was not effective (48.11 ± 5.68 vs. 47.61 ± 4.37 pg/mL).

### 3.4. Histopathological Examination of LiverTissue

#### 3.4.1. The Basic Lobular Structure 

The basic structure of the liver lobules observed under low power objective of Masson’s Trichrome stained specimens was fairly well preserved, only two Fructose animals reaching above minimal changes in semi-quantitative score grades (Figure 5a–d). In the transversal cross-sections, the lobules were filled by cord parenchymal cells, which radiated from the central vein and were separated by adjacent sinusoids without prominent signs of cord thickening. Sinusoidal irregularity and disruptions of tissue architecture in the Fructose group (presumably with regard to centrilobular fibrosis) did not surpass the mild value or displayed significant differences among groups (Figure 5e,f and Appendix A). 

#### 3.4.2. Glycogen Deposits

Glycogen deposits in parenchymal hepatocytes, visible as diffuse pigment or dark purple grains showing cell polarity, were documented in HE-stained specimens (Figure 6a–d). Figure 6b shows the clear zonation in increasing glycogen deposits in liver parenchyma around the portal area from the Fructose group, where fructose feeding allows for the depositing process to be the most potent. In contrast to this image intensity and regularity in the Fructose group, in the remaining groups, i.e., Control, Fructose + MH-76 and Fructose + Prazosin, HE staining showed rather fainter reactions for glycogen deposits. Moreover, hepatocytes still showing purple glycogen deposits were irregularly distributed in the lobule area, and the distribution of deposit-laden hepatocytes was variable in the area of tissue sections. 

#### 3.4.3. Steatosis

In our investigation, minimal to mild macrovesicular steatosis was found only in the Fructose group (Figure 7a,b); three other groups were classified below minimal score (Figure 7c upper panel) using HE-stained images, verified however by Masson’s Trichrome stain when glycogen deposits had blurred the steatosis readout. Ballooning cells, subsequently developed from macrovesicular cells, were very rarely scattered throughout the specimens; similarly, cells revealing the Mallory–Denk bodies were extremely scarce (Figure 7b,d and Appendix A). Mild to moderate microvesicular steatosis was found unevenly distributed in a panlobular pattern (still accentuated in perivenular areas) in the Fructose group. In the Fructose + Prazosin and Fructose + MH-76 groups, microsteasteatosis did not exceed mild level, and in the Control group was at minimal score, showing significant difference with the Fructose group (*p* > 0.001) (Figure 7c lower panel).

#### 3.4.4. Periportal Fibrosis and Inflammation

Collagen fiber deposits and leucocytic infiltration in the periportal area were assessed in Masson’s trichrome slides. The amount of fibrous tissue depends on the size of the portal spaces and is directly proportional to the size of the structures of the portal triad. Additionally, longitudinal cuts that cross the length of the specimen can suggest fibrous walls or bridges that do not really exist. Considering both those limitations, only the transversal cross-section, separately for large and small portal structures (150 micrometer in diameter set as arbitrary boundaries), were assessed. The portal tracts, both small and large, tended slightly to be enlarged by fibrosis in the Fructose + Prazosin group (Figure 8a–f); however, for large portal areas the periportal fibrosis was classified below mild score in this group and in comparison to the Control group showed no significant differences in collagen fibers deposits (Figure 8e). No signs of bridging fibrosis were seen. On the other hand, the small portal tracts were mildly to moderately infiltrated by leukocytes in the Fructose and Fructose + Prazosin groups, which revealed significant differences compared to the Control group (Figure 8a–d,h). However, no signs of further inflammation or hepatitis were prominent [37]. An analogous tendency of alteration was observed in large portal areas, although only the Fructose + Prazosin group reached the statistical significance of observed changes (Figure 8g). In all investigated groups, the periportal areas were populated by viable hepatocytes. Lobular zone 1 also did not show eminent inflammation, hepatocellular edema (ballooning) (Figure 7a,b), isolated necrosis or apoptosis dispersed in the lobule, acidophilic bodies, and accumulations of macrophages (Figure 9j and Appendix A).

#### 3.4.5. Pericentral Fibrosis and Inflammation

The thickness of the adventitia of the venular wall was determined on tissue sections stained with Masson’s trichrome stain for connective tissue and Sirius red stain for collagens. These assessments, due to staining specificity, may generally give off some discrepancy on the reported thickness of the central vein wall. No reference data on the wall thickness are available for rodents; the not-thickened wall is barely discernible at the light microscopic level [39]. Considering that as the venular diameter becomes greater, the wall thickness increases, small central veins and large central veins were assessed separately with an arbitrary boundary set at 100 micrometer diameter value (justifying this choice with the available data as the mean diameter of the central veins is about 80 μm [39]). The most common feature of perivenular fibrosis in Fructose livers was graded as mild to moderate, which was significantly higher than in Control and Fructose + MH-76-treated animals (Figure 9a–d,f). Rather, only fine pericellular and perisinusoidal deposits without significant differences among experimental groups were seen (Figure 9a–e,g); and using histochemical reaction for collagen, we could not determine their fibrogenic character [39]. In fibrotic perivenular regions, foci of leucocytic infiltration (Figure 7b) and aggregated Kupffer cells (Figure 9j and Appendix A) were seen.

Moreover, mild frequency grade facial apoptosis was observed in fibrotic perivenular areas of the Fructose and Fructose + Prazosin groups, and this observation was significantly more frequent than in Control as well as Fructose + MH-76 livers (Figure 9a–d,h). We also observed occlusion and obstruction changes in the Fructose group more frequently than in other groups (Figure 9a–d,i).

#### 3.4.6. Sinusoidal Changes and Sinusoidal Cellularity

In accordance with the fibrotic alteration and obstructive/occlusive changes in central veins, we observed tendency to dilation, occlusion, and congestion of sinusoids in all fructose-fed groups (Figure 10a,b). The significance of those observations was not confirmed statistically, presumably because of the great variability within specimens (Figure 10a–d). An increase in cells in the sinusoidal space can be a subtle finding that requires analysis with high magnification, and it was performed using 100× objective. Apart of locally accumulated macrophages in lobular zone 3, we observed increased sinusoidal cellularity in regards to neutrophils in fructose-fed groups (Figure 10e and Appendix A). 

#### 3.4.7. Pigment Deposits

Observations of pigment deposits (presumably iron deposits) were scattered and scarce in Control livers, showing minimal grade on average but displaying large deviation among other specimens, especially in the Fructose + MH-76 group. This resulted in that statistical differences in this variable could not be demonstrated. Nonetheless, Figure 11a shows a tendency towards alteration between the Control and the experimental groups. Hemochromatosis was also rarely identified throughout the lobules and in hepatocytes, Kupffer cells, portal stromal cells and ductal epithelia (Figure 11 and Appendix A).

### 3.5. Hepatotoxicity of MH-76 and Prazosin

The hepatotoxicity of MH-76 and prazosin was evaluated in vitro with use of the hepatoma HepG2 cell line. The statistically significant hepatotoxic effect was observed for both compounds only at highest used doses of 50 and 100 µM and was much weaker than that observed for the positive control doxorubicin (only around 10% of viability at 1 µM) (Figure 12a,b). Moreover, the MH-76 toxicity at 50 and 100 µM was less than that of prazosin (71.5 vs. 17.3% of viability at 50 µM and 9.5 vs. 0.8% of viability at 100 µM, respectively).

## 4. Discussion

Prevalence of metabolic syndrome, associated with excessive fructose consumption, has largely increased over the past decades mainly as a result of the high consumption of industrially manufactured foods in which fructose glucose syrup is added [2,4,40]. There is strong evidence that the fructose-rich diet may cause obesity, insulin resistance and MAFLD/NAFLD, a hepatic component of metabolic syndrome [1,40,41,42]. 

Fructose is an exceptionally lipogenic sugar. In the intestine already, there are no feedback mechanisms to suppress fructose absorption or transportation. The transcription of glucose transporter 5 (GLUT5) in the intestine increases due to fructose stimulation, which leads to further enhancement in fructose absorption [42]. After absorption, fructose undergoes the first-pass metabolism in the liver. The main metabolic pathway is fructolysis, which omits the step of phosphofructokinase, which limits the metabolic speed. Hepatic fructolysis is initiated by the phosphorylation of fructose into fructose-1-phosphate by ketohexokinase. Through aldolase B and triokinase activities, fructose-1-phosphate is then split into two triose phosphate intermediaries. Limitless fructolysis rapidly results in the availability of downstream (triose phosphate) intermediaries and is responsible for high levels of de novo lipogenesis as well as gluconeogenesis [40,43,44]. These processes lead to alteration of triglycerides metabolism and induce insulin resistance, steatosis, dyslipidemia and visceral adiposity [42,43]. 

In our experiment, 12 weeks of fructose feeding caused blood pressure elevation accompanied with increased fasting glycemia in rats, which indicated that metabolic syndrome had been induced (Appendix A). Since then, the treatment with MH-76 and prazosin started, with further fructose administration for a subsequent 6 weeks. At the end of the experimental period, the rats from the Fructose group presented hypertension, hyperglycemia, insulin resistance and abdominal adiposity with inflammation (Appendix A) [30,31]. We showed that both α1-adrenoceptor antagonists, MH-76 and prazosin, exerted hypotensive effects and reduced hyperglycemia; however, only MH-76 decreased insulin resistance, hypertriglyceridemia and reduced abdominal adiposity and adipose tissue inflammation (Appendix A) [30,31]. As a continuation of our previous research, in this study we aimed to investigate the effect of fructose consumption and treatment with α1-adrenoceptor antagonists of different classes on liver tissue in detail. 

In the histopathological analysis, the liver tissue of all groups appeared to be normal or almost normal when observed under low power objective. This is in line with observation of other researchers [45], who reported that in some cases, especially in entities that cause non-cirrhotic processes, portal and lobular changes are subtle and focal and can easily pass unnoticed. Therefore, we decided to perform a more detailed histopathological analysis and describe precisely morphological changes found, even though specific massive lesions were not identified in the specimens.

Steatosis is considered as the hepatic manifestation of the metabolic syndrome. The main liver cells are hepatocytes, in which triglycerides accumulate and the excessive lipid content in these cells is typical for NAFLD/MAFLD. Intrahepatic lipid accumulation may be the result of, in part, increased delivery of fatty acids to the liver from insulin-resistant adipose tissue and increased hepatic fatty acids synthesis from de novo lipogenesis. Hepatic fructose metabolism rapidly results in the availability of de novo lipogenesis substrates, which act as nutritional regulators of key transcription factors including carbohydrate response element-binding protein (ChREBP) and coactivators for genes involved in de novo lipogenesis. High fructose diets commonly induce systemic insulin resistance, hyperglycemia and fasting hyperinsulinemia, promoting another transcription factor for genes in the novo lipogenesis pathway: sterol regulatory element-binding protein 1c (SREPB1c). The above processes promote lipid deposition in liver, hepatic steatosis, which was confirmed both in human and rodent studies [40,46,47,48].

Hepatocellular steatosis is classified into two types: macrovesicular and microvesicular. In macrovesicular steatosis, a single large fat droplet or smaller well-defined fat droplets occupy the cytoplasm of hepatocytes, pushing the nucleus to the periphery. In microvesicular steatosis, the cytoplasm of hepatocytes is filled with tiny lipid droplets, and the nucleus is located centrally in the cell. Considering the consequences of both types of hepatocellular steatosis, microvesicular steatosis is considered more acute, and it can lead to life-threatening liver failure relatively quickly. In contrast, macrovesicular steatosis is often associated with chronic liver conditions, such as fibrosis and cirrhosis, which may progress slowly over time [49,50]. Steatosis in MAFLD/NAFLD is usually macrovesicular; however, microvesicular steatosis may also occur [50,51]. Steatosis in NAFLD usually begins in zone 3, although panlobular steatosis may also be seen with severe steatosis. In our investigation, minimal to mild macrovesicular steatosis was found only in the Fructose group; three other groups were classified below minimal score. This is in line with the results of triglycerides content in liver tissue, which was increased in the Fructose group but not in fructose-fed rats treated with both α1-adrenolytics. Microvesicular steatosis was more prominent in the Fructose and Fructose + Prazosin groups, and that pathology was alleviated with treatment with MH-76. It is known that, in condition of insulin resistance, hepatic de novo lipogenesis is activated [40,41,46,47,48], leading to fat accumulation in the liver. We previously showed that MH-76 but not prazosin reduced hyperinsulinemia and insulin resistance [30,31], and this may explain the higher ability of MH-76 than prazosin to reduce steatosis.

In our histopathological analysis, we showed the most prominent glycogen deposits in livers from the Fructose group and the zonation on glycogen deposits in liver parenchyma of Fructose animals. A large influx of fructose into the liver causes accumulation of glycogen (via gluconeogenesis) apart of triglycerides [40]. Hepatic glycogen storage also stimulates de novo lipogenesis [42] leading subsequently to reduced insulin sensitivity, insulin resistance and glucose intolerance featured in this model [52]. Treatment with MH-76 and prazosin exerted beneficial effects, decreasing the intensity of glycogen deposits. 

In our study, we found that fructose overconsumption led to elevated leptin concentration in liver tissue. Leptin controls fat catabolism and glucose production in liver tissue. Increasing lipogenesis in the liver due to fructose overfeeding leads to the intracellular accumulation of malonyl-CoA, which represents a disturbed balance between synthesis from acetyl-CoA and utilization in fatty acid synthesis and degradation to acetyl-CoA. An excess of malonyl-CoA leads to abnormal production of leptin and inhibits hepatic lipid β-oxidation [43]. The reduction in hepatic fatty acid oxidation in fructose-fed rats is attributable to incomplete activation by leptin of two proteins involved in the control of fatty acid catabolism: the enzyme AMPK and the peroxisome proliferator-activated receptor α (PPARα). It was also shown that fructose administration to rats resulted in hyperleptinemia and hepatic leptin resistance. This was caused by impairment of the leptin-signal transduction mediated by both janus-activated kinase-2 and the mitogen-activated protein kinase pathway, leading finally to hypertriglyceridemia and hepatic steatosis [53]. Treatment with MH-76 reduced the increased leptin concentration, whereas prazosin was not effective. A similar observation was made in adipose tissue of the same animals; we found higher leptin concentration, which was reduced by treatment with MH-76 but not with prazosin [31]. 

Insulin resistance, hyperleptinemia, increased hepatic lipogenesis and hyperinsulinemia result in elevated toxic metabolites which can act as reactive oxygen species. Fructose directly and indirectly facilitates oxidative damage and lipid peroxidation, a process in which unsaturated lipids become oxidatively degraded to a variety of products at sites of inflammation [6,44]. These effects create a lipotoxic environment for the hepatocytes. The lipotoxic oxidative stress results in hepatocellular mitochondrial dysfunction and endoplasmic reticulum stress that enhance the oxidative stress and downregulate the nuclear receptor PPARα—a major transcription factor involved in the regulation of fatty acid oxidation. These events result in hepatocellular inflammation, apoptosis and finally, hepatic fibrosis [6,43]. In our previous study, we showed that fructose feeding markedly increased lipid peroxidation in liver, whereas treatment with both α1-adrenolytics decreased lipid peroxidation [30]. In this study, we observed that fructose feeding caused a significant decrease in NPSH in liver. The sum of low molecular weight thiols (in reduced form), such as glutathione (GSH), homocysteine, cysteine and cysteinylglycine, is referred to as non-protein sulfhydryl groups (NPSH) [54]. NPSH are potent reducing agents, and as important antioxidants, play a role in the detoxification of a variety of electrophilic compounds, such as reactive oxygen species/reactive nitrogen species (ROS/RNS) scavengers, to prevent them from oxidizing proteins, lipids and DNA [55]. Decreased level of GSH causes higher ROS production, which results in an imbalanced immune response and inflammation. GSH depletion may cause mitochondrial dysfunction, which plays an important role in the process of apoptosis. The imbalance of cellular ROS and GSH may lead to inhibition of cell growth and proliferation and result in cell death [56]. We have shown that administration of the compound MH-76 as well as prazosin leads to recovery of the non-protein thiol pool in the liver and significant increase of the reducing power in liver tissue. 

Histopathological analysis showed also mild to moderate fibrosis with leucocyte infiltration in periportal and pericentral areas in the Fructose and Fructose + Prazosin groups. In accordance with these results, in most previous research prominent portal fibrosis is not an immanent characteristic of metabolic syndrome, and portal inflammation is usually absent or mild and consists mainly of lymphocytes [37]. Fibrosis of the central veins is a characteristic histological lesion found in alcoholic patients. In studies by Worner and Lieber (1985) [57] of young adult heavy alcohol drinkers, fibrosis of the central vein was defined by the thickening of the venular wall, which was invariably accompanied by band-like fibrous strands spreading into the perivenous parenchyma. In MAFLD, the further characteristic is not always developed [39]. Accordingly, such a “chicken wire” perivenular fibrosis was observed only scarcely. Rather, only fine pericellular and perisinusoidal deposits without significant differences among experimental groups were seen. However, these observations, only rarely attracting attention in histopathological assessments, may have further consequences. Perivenular fibrosis can progress to occlusion of the central vein [39]. The perivenous hepatocytes are proposed to act as stem/progenitor cells to provoke hepatic homeostatic cell renewal. Central vein fibrosis and perivenous perisinusoidal fibrosis may affect perivenous gene expression. Fibrotic veins containing increasing extracellular matrix deposits could increase the endothelium–hepatocyte proximity, potentially disrupting contacts between the central vein endothelium and surrounding perivenous hepatocytes. This may alter the local environment of perivenous hepatocytes, which in turn could affect the transit and/or strength of endothelium-derived Wnt signals [39], the master regulator of hepatic metabolic zonation in rodents. In fibrotic perivenular regions, foci of leucocytic infiltration were seen. Nevertheless, the most significant fibrotic lesions accompanied with leucocytic infiltration was found in the Fructose and Fructose + Prazosin groups but not in the Fructose + MH-76 group. Our biochemical assays showed also that mild inflammation was present in the liver of fructose-fed rats, indicated by elevated levels of IL-6 in liver tissue. Other parameters of inflammation were not elevated in the liver, but our previous studies show that the level of other cytokines—TNF-α and MCP-1—increased in the adipose tissue of the same rats [30]. Considering the fact that inflammatory and anti-inflammatory factors are released sequentially [58] and IL-6 down-regulates the synthesis of IL-1 and TNF [59,60], therefore only selected inflammatory cytokines may be detectable in certain time windows [58]. It is established that pro-inflammatory cytokines, including IL-6, aggravate the oxidative damage of the vital organs, particularly the liver [61,62]. IL-6 may be also involved in the pathogenesis of hepatic insulin resistance as insulin sensitivity increases in diet-induced obese mice treated with anti-IL-6 antibodies [3]. Thus, the results obtained by us that the administration of MH-76 significantly reduces the level of IL-6 in the liver shows one of the mechanisms of beneficial effects of this compound on hepatocytes of fructose-fed rats. On the contrary, prazosin treatment did not reduce elevated IL-6 concentration. Similar results were found in our previous studies; treatment with prazosin did not reduce the elevated concentrations of leptin, TNF-α, IL-6 and MCP-1, and did not improve insulin signaling in inflamed adipose tissue from fructose-fed rats [30,31]. We also assessed the concentration of plasminogen activator inhibitor 1 (PAI-1), an acute-phase protein which is a key modulator of hepatic lipid transport. PAI-1, in later stages of the disease (steatohepatitis), contributes also to inflammation. Increased PAI-1 concentrations have been linked to liver fibrosis but also to earlier stages, e.g., steatosis of NAFLD [42,63]. However, in our studies we did not observe higher hepatic PAI-1 concentration in fructose-fed rats. This may be due to the fact that the observed liver inflammatory lesions were rather mild.

Apart from inflammation, surpassing mild frequency grade facial apoptosis was observed in fibrotic perivenular areas of the Fructose and Fructose + Prazosine groups, but not in the Fructose + MH-76 group. It is known that hepatocytes in lipotoxic states caused by fructose may undergo apoptosis [64]. On the other hand, numerous studies showed that quinazoline-based compounds like doxazosin, prazosin or their derivatives may exert apoptosis [20,21]. In our study, we showed that treatment with MH-76 but not prazosin reduced the occurrence of apoptosis in the perivenular area. 

Histopathological analysis revealed also a tendency to dilation, occlusion and congestion of sinusoids in fructose-fed rats accompanied with neutrophil infiltration. These changes were not attenuated by studied compounds. Neutrophil-mediated liver injury has been reported in various types of liver diseases including metabolic-associated fatty liver disease and drug-induced liver injury [39]. 

Liver conditions such as metabolic-associated fatty liver disease can cause hemosiderosis [45]. It is characterized by brown-colored granular pigment deposits in periportal areas. In our study, we showed that pigment deposits were present in all groups of fructose-fed rats. Hemochromatosis was also rarely identified throughout the lobules, in hepatocytes, Kupffer cells, portal stromal cells and ductal epithelia. 

In summary, histopathological analysis showed that fructose-rich diet in rats promotes hepatocellular steatosis and tissue inflammation. However, the hepatic phenotype of steatohepatitis (accordingly to [36,37]) may not be produced by using a high fructose diet alone. Widely accepted scoring systems for staging of chronic hepatitis in NAFLD using sum of scores for steatosis (5–33% of hepatocytes involved, score1), lobular inflammation (2 foci per 200× field, score1) and hepatocellular ballooning (few ballooned cells, score 1) reach value 3, which is at the lowest value of uncertain hepatitis [36]. Necessary components of histopathological abnormalities in NAFLD/metabolic-associated steatohepatitis (although a complete consensus has not been reached) are steatosis (macro > micro; accentuated in zone 3), lobular inflammation (mixed, mild; scattered polymorphonuclear leukocytes as well as mononuclear cells) and hepatocellular ballooning (most apparent near steatotic liver cells, typically in zone 3). Usually present but not necessary for diagnosis are perisinusoidal fibrosis (in zone 3), hepatocellular glycogenated nuclei (in zone 1), lipogranulomas (in the lobules; of varying size, but usually small) and acidophil bodies or periodic acid-Schiff-stained Kupffer cells. Finally, abnormalities that may be present but not necessary for diagnosis are Mallory-Denk bodies (in ballooned hepatocytes), iron deposition (in hepatocytes or sinusoidal lining cells) and megamitochondria in hepatocytes [37]. All those observations were made in tissue slides, some of them however too accidentally. An improved understanding of hepatic phenotype produced by high fructose diet gives the observation of fibrotic central vein wall and associated progressive pericellular and perisinusoidal fibrosis, which may comprise integrating the structures, biology and physiology of liver sinusoids and central veins in mediating metabolic homeostasis and hepatic regeneration. Nevertheless, we showed that fructose-fed rats developed changes typical for MAFLD such as higher density of glycogen deposits in parenchymal hepatocytes, macro- and microvesicular steatosis, mild periportal fibrosis with leucocyte infiltration, fibrosis in small centrilobular regions, signs of occlusions/obstructions of central veins, apoptosis in perivenular aeras and neutrophils in lumen of the portal vein. In addition, steatosis was confirmed in biochemical assays, which showed higher triglycerides concentration in livers from the Fructose group, and mild inflammation found in histopathological analysis was confirmed with higher IL-6 level and accompanied with lower antioxidant capacity in livers from the Fructose group. These abnormalities were alleviated to some extent by treatment with α1-adrenolytics, with MH-76 being more advantageous due to its anti-inflammatory effect as well as an ability to reduce insulin resistance [30,31]. Recently, it has been shown that α1-adrenoceptor antagonists used earlier to treat benign prostatic hyperplasia (BPH), post-traumatic stress disorder (PTSD) or arterial hypertension may be effective in reducing mortality related to COVID-19-associated hyperinflammation [28]. Patients with confirmed COVID-19 taking any of the α1-adrenoceptor antagonists (tamsulosin, terazosin, prazosin, doxazosin, alfuzosin, silodosin), compared to non-users, had a 20% relative risk reduction for death. Similar results were obtained in patients with ARDS or pneumonia, suggesting that the benefits of α_1_-adrenoceptor inhibition for mortality may be independent of etiology in patients with lower respiratory tract infection or inflammation. Among different α1-adrenceptor antagonists, doxazosin was associated with the highest relative risk reduction for death, while tamsulosin had a more modest relative risk reduction, suggesting that blockade of all α1-adrenoceptor subtypes is more beneficial regarding immunomodulatory properties of α1-adrenoceptor antagonists. As well, pre-clinical data suggest also that non-selective α_1_-adrenoceptor antagonists are effective in preventing hyperinflammation and death in animal models of cytokine storm syndrome and may exert anti-inflammatory and immunomodulatory properties [28]. Our study also shows that α_1_-adrenoceptor antagonists may exert anti-inflammatory effects and that α_1_-adrenolytics are a heterogeneous group of drugs. We compared two non-subtype selective α_1_-adrenoceptor antagonists with different chemical structure, and MH-76 turn out to be more beneficial than prazosin regarding alleviation of metabolic syndrome and exerting additional anti-inflammatory effect in adipose [31] as well as hepatic tissues. 

When comparing the effects of prazosin and MH-76, their influence on the sympathetic nervous system should be considered. Sympathetic overactivation has also been shown to take a role in the development of metabolic syndrome. There is strong evidence implicating sympathetic overactivation in the pathogenesis of obesity, diabetes, hypertension and MAFLD [6,13]. Increased sympathetic activity mediates insulin resistance, which in turn leads to compensatory hyperinsulinemia and hyperglycemia and progression of MAFLD. Direct sympathetic stimulation induces peripheral vasoconstriction in skeletal muscles, which results in impaired glucose uptake, and increases glycogenolyses and gluconeogenesis in liver and glucose reabsorption and sodium retention in kidneys, among others [6,13]. Therefore, α_1_-adrenoceptor antagonists may counteract some of these effects, and by lowering peripheral resistance improve glucose uptake and reduce hyperglycemia. However, prazosin administration may be accompanied with reflex tachycardia due to unfavorable pharmacokinetic properties. In our previous study, we found that chronic administration of prazosin lowered blood pressure but not heart rate, and we found an increased norepinephrine concentration following prazosin treatment [65]. On the contrary, treatment with MH-76 decreased heart rate and did not alter norepinephrine serum concentration. MH-76, a 1-(2-methoxyphenyl)piperazine derivative, may be regarded as a urapidil derivative, and similarly to urapidil, it is also a 5-HT1A serotoninergic receptor ligand [65]. It is possible that MH-76 also reduces the sympathetic activation by acting on central 5-HT1A receptors, which may be associated with more favorable effects on metabolic abnormalities in fructose-fed rats. 

In the discussion presented above, the efficacy of both investigated compounds was directly compared to each other; however, some additional aspects influencing their pharmacological activities also need to be considered such as differences in the affinity for α_1_-receptors, and pharmacokinetic properties, mainly liver distribution. The differences in the receptor affinity were compensated by an administration of different doses. Since the K_i_ for prazosin was ca. 20 times lower compared to K_i_ obtained for compound MH-76, the tested dose of MH-76 was appropriately higher [31]. In terms of liver distribution, we directly measured the concentrations of investigated drugs in the liver at the end of the experiment. Although prazosin was administered at a lower dose compared to MH-76, its concentration in the liver was much higher (301.32 ± 171.46 ng/g vs. 101.07 ± 36.13 ng/g). Such results were of no surprise since prazosin is extensively metabolized in the liver and only 6% of this drug is excreted unchanged, mainly in the urine [66]. As for the compound MH-76, considering its higher administered dose, the amount of drug in the liver was rather low [31]. Since the antagonistic potency of prazosin towards α1-receptors is two times higher compared to MH-76 (0.68 nM vs. 1.42 nM), there are probably mechanisms other than α1-adrenoceptors antagonism involved in the activity of MH-76 observed in liver tissue. 

In the end, we also evaluated and compared direct hepatotoxic effects of prazosin and MH-76. The hepatotoxicity assay showed that MH-76 was less toxic on hepatocytes than prazosin, which may be attributed to its chemical structure, as MH-76 did not contain quinazoline moiety. Quinazoline-based α_1_-adrenoceptor antagonists can induce apoptosis in various normal, benign and malignant cells. Such apoptosis-inducing effects occur independently of α_1_-adrenoceptor antagonism and were never shown for non-quinazoline α_1_-adrenoceptor antagonists such as tamsulosin or urapidil [21,22,67]. 

One of the most important limitations of our study is that we did not assess the proinflammatory cytokines’ concentration in the blood; we determined them only locally, in the liver tissue. So we are not able to conclude whether the cytokines’ tissue concentration was correlated with their level in the blood, which would be valuable from the clinical point of view. However, in our basic studies we focused on local concentration of cytokines, as it may be more relevant in study concerning the pathogenesis of the metabolic syndrome.

Summing up, we showed that fructose administration has detrimental effects on liver tissue in rats inducing steatosis and mild inflammation, and some of these changes were alleviated by MH-76 and to a lesser extent by prazosin. α1-adrenoceptor antagonists may still be a valuable group of antihypertensive drugs as they may improve fructose-induced metabolic impairment, along with blood pressure reduction; however, it should be considered that this is a heterogeneous group of drugs, with different chemical structure and pleiotropic effects.

## Data Availability

The data presented in this study are available on request from the corresponding author. Data is not publicly available due to privacy or ethical restrictions.

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
