# Peer review of "Manifestations of Liver Impairment and the Effects of MH-76, a Non-Quinazoline α1-Adrenoceptor Antagonist, and Prazosin on Liver Tissue in Fructose-Induced Metabolic Syndrome"

_metabolites, 2023, doi:10.3390/metabo13111130_

Round 1

Reviewer 1 Report

Comments and Suggestions for Authors

I have read and analyzed the manuscript from Kubacka and coauthors. In my opinion, the suggested study is interesting but has some problems in part with figures organization and introduction motivation. However, I would like that the discussion  can help authors to  improve the manuscript for further publication consideration.

  1. Authors devote their manuscrit to the action of alpha1 adrenoreceptor antagonist MH76 on liver function and fibrosis. How is liver injury related to alpha1 adrenoreceptor function? It is not clarified in the introduction clearly.

  2. I see this point of view in some manuscripts, but in my opinion every study with animal model should contain model characterization, at least as a supplement, but not just as a reference. Which criteria were used for the metabolic syndrome diagnosis in animal model? Did authors measure blood pressure in rats? How does MH76 change blood pressure? Did authors evaluate insulin sensitivity?

  3. Authors wrote that fructose supplementation was performed by addition to water. How did authors control fructose consumption? 

  4. Why did authors evaluate TNF, IL6 and other cytokines concentration in the liver, but not in blood?

  5. Statistical analysis. Why did authors in the single study use both parametric and non parametric statistics?

  6. 20 figures are not appropriate for an experimental manuscript. Authors must reduce figures number up to 10 as a maximal. In my opinion authors can conflate Figures 3-6, 7-9, 11-16.

  7. Figures 8, 10, 12, 15, 17, 19. All of them should contain representative images for every group.

  8. Figure 9. IHC is not a relevant method for the glycogen accumulation estimation. Moreover, all conclusions about immune cells accumulation can not be made without immunofluorescent staining in immune cells markers (specific markers for macrophages, neutrophils and others).

  9. Figures 11 and 14. Author should quantitatively evaluate Sirius red staining, just representative images are insufficient. Moreover, leukocyte infiltration can not be shown without specific markers.

Thus, in my opinion, the manuscript can be considered for publication but just after intensive review and modifications.

Reviewer 2 Report

Comments and Suggestions for Authors

Kubacka et. al conducted a study titled “Manifestations of liver impairment and the effects of MH-76, a non-quinazoline α1-adrenoceptor antagonist and prazosin on liver tissue in fructose-induced metabolic syndrome”. This study demonstrated α1-adrenoceptor antagonists MH-76 may an interesting option for treatment of MAFLD. However, there are a few points/ suggestions for further clarification or comments from the authors.

1.     The design of the animal experiment is problematic. The author administered fructose water to both the fructose group and the treatment groups for 12 weeks, and then switched to saline water for the subsequent 6 weeks. However, there is a self-recovering process occurring during the last 6 weeks. The author should have continuously provided fructose water for 18 weeks and administered treatments on the basis of the fructose water.

2.     The author also needs to provide some very crucial data: serum ALT level, AST level, and glucose level.

3.     The pathological section of the liver tissues in fructose and treatment groups show very minor differences, making it hard to discern any significant variations.

Comments on the Quality of English Language

The Quality of English Language is fine. 

Round 2

Reviewer 1 Report

Comments and Suggestions for Authors

Many thanks to the authors for the comprehensive response. I have just a couple of small corrections.

1.Ok.

2.Ok.

3.Ok, but can authors show the graph of fructose solution consumption? 

4.I agree with the authors, but tissue cytokines measurement has less clinical value than blood measurement. It should be reflected in Limitations.

5.What about the Shapiro Wilk test application? 

6.Ok.

7.Ok.

8.Ok.

Reviewer 2 Report

Comments and Suggestions for Authors

The authors have adequately addressed all of my concerns. I have no further comments.

Author Response

Thank you again for your review.